# SCBench: A Testbed for Causal Inference with Time Series Panel Data

Megan Richards [1]  Saeyoung Rho [2]  Kyunghyun Cho [1]

## Abstract

Synthetic control (SC) is a widely used method for estimating causal effects from observational panel data, with classical approaches expressing a target trajectory as a linear combination of donor trajectories. Recent advances in foundation models pretrained on synthetic data have shown strong performance in structured, sample-limited regimes such as tabular and time-series prediction, raising the question of whether such models are also effective for synthetic control. We conduct a large-scale empirical study of traditional and foundation model approaches on over 300K simulated panels, spanning fully linear to fully nonlinear state-space dynamics, varying noise regimes, and a range of panel sizes and ranks. Comparing three foundation models (TabPFN, TabPFN-TS, and Chronos) against standard SC baselines (Robust SC, Lasso, and Simplex), we identify regimes where foundation models outperform classical baselines, including when latent dynamics are nonlinear and panels are high rank. Linear methods remain competitive or superior in low-rank and near-linear settings, with Simplex providing a reliable baseline across our testbed. These results suggest that foundation models pretrained on synthetic data are a promising direction for synthetic control in challenging regimes.

## 1. Introduction

Synthetic control (SC) is a widely used method for estimating causal effects from observational time-series data. Synthetic control was developed for use in economic and policy, where many questions of interest lack the controls required for direct effect estimation. For example, say that the state of California implements a new tax with the intention of reducing smoking rates (as in (Abadie et al., 2010)). Estimating the effect of the tax on smoking rates is nontrivial

[1]New York University [2]Columbia University. Correspondence to: Megan Richards <mr7401@nyu.edu>.

*Proceedings of the $2^{nd}$ ICML Workshop on Foundation Models for Structured Data*, Seoul, South Korea. 2026. Copyright 2026 by the author(s).

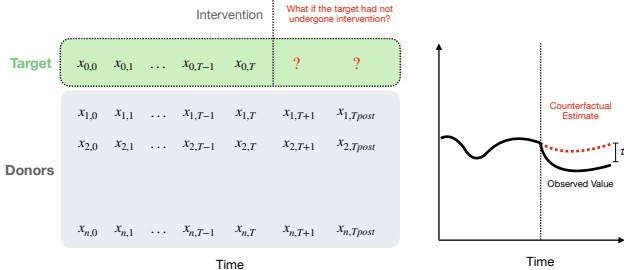

*Figure 1.* The synthetic control task is to predict the future time steps of a target using trajectories of a set of donors (left). The treatment effect is then estimated by comparing the predictions to the observed values under intervention (right).

due to the absence of a natural control (i.e., a very similar state, which did not introduce the policy). This challenge arises broadly in evaluating both natural interventions, such as extreme weather events, and applied interventions, such as new policies and regulations.

Synthetic control addresses this by constructing a control from a set of candidate units. The intuition is that while no individual state may serve as a faithful control, a combination of states may. In the state example, SC first identifies a linear combination of states ('donors') whose pre-intervention smoking rates closely track California's smoking rates. Assuming this relationship would have persisted absent the intervention, the same combination is used to predict the counterfactual post-intervention smoking rate for California. The causal effect is then estimated as the gap between this counterfactual and the observed trajectory.

The problem of synthetic control is uniquely structured. Each problem instance is a panel of time-series trajectories (Figure 1). Rather than extrapolating a single series, the goal is to predict the future time-steps of one trajectory (the target) from the trajectories of others (the donors). Therefore, the core inductive task is comparison across trajectories within a set. The predominant methods for synthetic control are linear, expressing the target as a weighted sum of donors. Recent work has shown that large models trained on synthetic data can be highly effective in structured, sample-limited regimes such as tabular data and time-series forecasting, suggesting opportunities for use in synthetic control.

In this work, we conduct a large-scale empirical study of traditional and foundation model approaches on over 300K simulated panels, spanning fully linear to fully nonlinear state-space dynamics, varying input noise regimes, and a range of panel sizes and ranks. We compare three representative foundation models (TabPFN, TabPFN-TS, and Chronos) against standard SC baselines (Robust SC, Lasso, and Simplex), and identify areas of substantial promise for foundation models on this task. In aggregate, TabPFN achieves the lowest mean and median error across our testbed, and foundation models more broadly excel in challenging regimes: they outperform traditional methods when latent dynamics are nonlinear and when low-rank assumptions do not hold, while exhibiting similar noise robustness to SC baselines. At the same time, linear methods remain competitive or superior in low-rank and near-linear settings, and provide a reliable baseline across the full range of conditions we study. Together, these results suggest that foundation models pretrained on synthetic data are a promising direction for synthetic control in challenging regimes. We provide a detailed discussion of opportunities for further study of these methods.

## 2. Related Work

**Synthetic control.** The synthetic control (SC) framework of (Abadie et al., 2010) estimates a counterfactual for a treated unit by reweighting donor units to match the pre-intervention outcome trajectory. SC has since become a standard tool in applied econometrics and social sciences (Abadie, 2021), with wide-ranging applications including in evaluating disaster response (Heersink et al., 2017), public health (Pieters et al., 2016), immigration enforcement (Bohn et al., 2014), and the efficacy of nonprofit programs (Abadie, 2021). Method development in SC has traditionally focused on improving the robustness of linear methods, including pre-processing steps such as RSC and weight regularizations such as Lasso and Simplex. More recent work has working to relax the assumption that donor-target relationships are constant over time (Rho et al., 2026), which exhibits better performance than linear methods when strong temporal trends exist.

**Foundation models for structured data.** Recent foundation models for time-series forecasting and tabular data have been developed by training large transformers on broad corpora of real and synthetic series. TabPFN (Hollmann et al., 2023) pretrains a transformer on millions of synthetic tabular datasets sampled from a structural prior, and at inference time produces predictions for a new dataset in a single forward pass. TabPFN-TS (Hoo et al., 2025) leverages TabPFN-v2 for forecasting problems by combining it with lightweight time-features, achieving strong performance on forecasting tasks without any additional training. (Das et al.,

| Parameter | Values |
|---|---|
| $\alpha = \beta$ (linearity) | {0, 0.25, 0.5, 0.75, 1} |
| Transition matrix | ortho ($\rho$=1), rescaled ($\rho \in \{0.8, 1\}$) |
| Loadings ($H, C$) | $\text{Dir}_5$, Gauss |
| Initial state ($x_0$) | $\text{Dir}_5$, Gauss |
| Noise covariance | diag, Wishart-$(d+1)$, Wishart-$10d$ |
| Noise level | low, high |
| Latent dim. $d$ | {2, 5, 10, 20} |
| Donors $n$ | {5, 10, 20, 50} |
| Pre-period $T_{\text{pre}}$ | {5, 10, 20, 50} |

*Fixed:* $T_{\text{post}} = T_{\text{pre}}$, burn-in=5, no normalization.

*Table 1.* Data generation parameter sweep. Each of the 16,200 parameter combinations is sampled 20 times, yielding 324,000 panels.

2024) introduce a decoder-only patched-attention model whose zero-shot accuracy approaches that of supervised baselines across a range of public datasets, and Chronos (Ansari et al., 2025) extend this paradigm to multivariate and covariate-informed forecasting via a group attention mechanism trained on synthetically structured multivariate data. A concurrent line of work (Illick et al., 2026) explores appraoches for adapting time-series foundation models for SC in a limited set of synthetic and real-world regimes, and proposes input representations tailored to the SC setting. We provide a complementary view at substantially larger scale, systematically varying linearity, noise structure, and panel rank across over 300K simulated panels.

### 2.1. Data Generation Process

We sample tasks from a nonlinear state-space model with $d$-dimensional latent state $x_t \in \mathbb{R}^d$ and $(n+1)$-dimensional observation $y_t \in \mathbb{R}^{n+1}$, where index 0 denotes the target unit and indices $1, \ldots, n$ denote donors. Each task draws fresh parameters $(A, B, H, C, Q, R, x_0)$ from the priors below:

$$x_t = (1 - \alpha) A x_{t-1} + \alpha \tanh(B x_{t-1}) + q_{t-1}, \quad (1)$$

$$y_t = (1 - \beta) H x_t + \beta \tanh(C x_t) + r_t, \quad (2)$$

where $q_{t-1} \sim \mathcal{N}(\mathbf{0}, Q)$ and $r_t \sim \mathcal{N}(\mathbf{0}, R)$.

We generate a large sweep of hyperparameters, summarized in Table 1 (see Appendix A.1 for full details). In total, we consider over 16K parameter combinations, with 20 independent samples for each, yielding 324K total panels.

## 3. Methods

### 3.1. Predictors

We compare traditional synthetic control methods, which express the target as a linear combination of donors, with foundation models pretrained on large corpora of synthetic or real data.

*Table 2.* We find that foundation models excel in nonlinear and high-rank data regimes, while linear methods outperform them in linear and low-rank data regimes. Shown is the median MSE compared to clean target values $h \leq 5$. The best baseline per row in **bold**; 95% CIs in grey. LASSO = CV-selected; RSC = silhouette-selected (best variant per family across settings; see Table 5).

| | Simplex | Lasso | RSC | TabPFN | TabPFN-TS | Chronos |
|---|---|---|---|---|---|---|
| *Linearity coefficient ($\alpha = \beta$)* | | | | | | |
| 0.00 | **1.08** (1.06, 1.10) | 1.44 (1.41, 1.47) | 1.12 (1.10, 1.15) | 1.18 (1.16, 1.21) | 1.30 (1.27, 1.32) | 1.43 (1.41, 1.46) |
| 0.25 | **0.60** (0.59, 0.61) | 0.90 (0.88, 0.91) | 0.66 (0.65, 0.67) | 0.60 (0.59, 0.61) | 0.68 (0.67, 0.69) | 0.67 (0.66, 0.69) |
| 0.50 | 0.48 (0.47, 0.49) | 0.70 (0.69, 0.71) | 0.67 (0.65, 0.68) | **0.43** (0.42, 0.44) | 0.49 (0.48, 0.50) | 0.47 (0.46, 0.48) |
| 0.75 | 0.43 (0.43, 0.44) | 0.59 (0.58, 0.60) | 0.72 (0.70, 0.73) | **0.36** (0.36, 0.37) | 0.42 (0.41, 0.42) | 0.38 (0.38, 0.39) |
| 1.00 | 0.48 (0.47, 0.49) | 0.62 (0.61, 0.63) | 0.78 (0.77, 0.80) | **0.41** (0.40, 0.41) | 0.46 (0.46, 0.47) | 0.43 (0.42, 0.44) |
| *Relative rank $d / \min(n, T)$* | | | | | | |
| 0.04 | 0.27 (0.25, 0.29) | 0.44 (0.42, 0.46) | **0.15** (0.14, 0.16) | 0.34 (0.33, 0.36) | 0.35 (0.33, 0.36) | 0.33 (0.31, 0.34) |
| 0.1 | 0.37 (0.36, 0.38) | 0.56 (0.55, 0.58) | **0.29** (0.28, 0.30) | 0.35 (0.34, 0.36) | 0.37 (0.36, 0.38) | 0.41 (0.39, 0.41) |
| 0.2 | 0.49 (0.48, 0.50) | 0.74 (0.73, 0.76) | 0.53 (0.52, 0.54) | **0.48** (0.47, 0.49) | 0.55 (0.54, 0.56) | 0.55 (0.54, 0.56) |
| 0.25 | 0.43 (0.42, 0.45) | 0.60 (0.59, 0.62) | 0.51 (0.49, 0.52) | **0.38** (0.37, 0.39) | 0.43 (0.42, 0.45) | 0.42 (0.41, 0.44) |
| 0.4 | **0.64** (0.62, 0.65) | 0.91 (0.90, 0.93) | 0.97 (0.95, 0.99) | **0.64** (0.62, 0.65) | 0.75 (0.74, 0.77) | 0.73 (0.72, 0.75) |
| 0.5 | 0.56 (0.55, 0.57) | 0.76 (0.75, 0.78) | 0.83 (0.82, 0.85) | **0.49** (0.48, 0.50) | 0.54 (0.53, 0.55) | 0.52 (0.51, 0.53) |
| 1 | 0.68 (0.67, 0.69) | 0.90 (0.89, 0.91) | 1.15 (1.13, 1.16) | **0.57** (0.56, 0.58) | 0.66 (0.65, 0.67) | 0.60 (0.60, 0.61) |

**Traditional baselines.** Simplex (Abadie et al., 2010) restricts donor weights to the unit simplex, requiring weights to be non-negative and sum to one. Lasso fits unconstrained weights with an $\ell_1$ penalty, inducing sparsity over donors. Robust SC (RSC) (Amjad et al., 2018) first denoises the donor matrix via singular-value thresholding, then fits unconstrained weights on the resulting low-rank approximation.

**Foundation models.** TabPFN (Hollmann et al., 2023) is a transformer pretrained on synthetic tabular regression tasks that performs in-context prediction without gradient updates. We apply it to SC by treating each donor unit as a feature and each time step as a row, with the target trajectory as the regression target. TabPFN-TS (Hoo et al., 2025) adapts TabPFN to time-series forecasting by augmenting each row with temporal features (e.g., position, seasonality encodings). Chronos (Ansari et al., 2025) is a transformer pretrained on a large corpus of real and synthetic time series. We apply it in the panel setting by passing donor trajectories as known future covariates alongside the target's pre-intervention history.

### 3.2. Evaluation

All methods are fit on pre-intervention time points ($T_{\text{pre}}$), and applied on the next $T_{\text{post}}$ steps for the target unit. For each task, we generate $T$ pre-intervention time steps (used as context) and an additional $T_{\text{post}}$ post-intervention time steps (used for evaluation), where $T_{\text{post}} \leq T$. We compare predictions against two ground-truth targets: noisy observations (with $r_t$) and the noiseless signal (without $r_t$), and evaluate predictions at $T_{post} = [1..5]$.

## 4. Results

We compare foundation models and traditional methods on our 300K synthetic panel testbed and find both areas of substantial promise for foundation models and regimes where linear methods remain the stronger choice. Table 2 summarizes our results, broken down by linearity coefficient and relative panel rank. Across all settings, TabPFN achieves the strongest mean and median performance overall, while Simplex remains the most reliable linear baseline. In the following sections, we characterize this tradeoff more closely. We first study how each method behaves as latent dynamics shift from linear to nonlinear (Section 4.1), then examine robustness under varying noise magnitudes and correlation structures (Section 4.2), and finally evaluate scaling with respect to panel rank (Section 4.3). Appendix B and B.7 contain full ablation results, plots broken down by additional axes, and all results replicated against noisy targets.

### 4.1. Performance Across Linearity Regimes

Figure 2 shows results as a function of the linearity coefficient, where $\alpha = \beta = 0$ reproduces a fully linear state-space model and $\alpha = \beta = 1$ produces the fully nonlinear model in Equation 2. Fully linear settings yield larger and more skewed errors across all methods, and the tanh nonlinearity appears to provide implicit regularization at higher $\alpha, \beta$. TabPFN achieves the lowest median MSE at all configurations with $\alpha = \beta \geq 0.50$, while Simplex matches or outperforms TabPFN in fully linear and near-linear settings.

### 4.2. Robustness to Noise

We evaluate each method under two noise magnitudes (low and high) and two correlation structures, one fully ran-

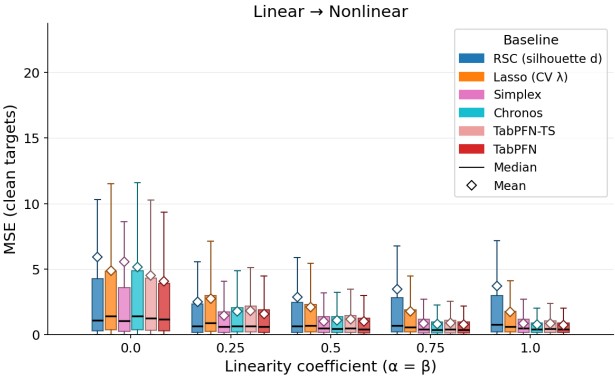

Figure 2. Testbed results according to linearity coefficients. $\alpha = \beta = 0$ reproduces a fully linear state-space model, and $\alpha = \beta = 1$ produces the fully nonlinear state-space model described in Eq. 2.

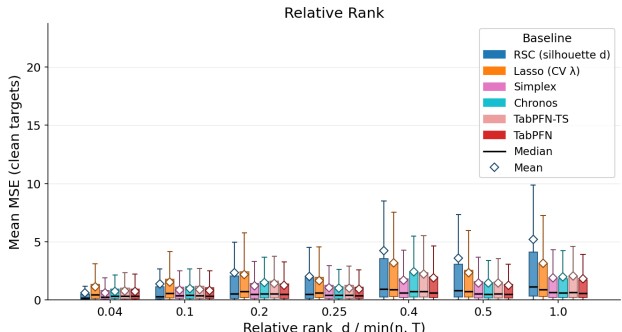

Figure 3. Testbed results as a function of relative rank, defined as the panel's rank divided by the minimum of the sample and time dimensions.

| Method | high, correlated | high, random | low, correlated | low, random |
|---|---|---|---|---|
| Simplex | 1.490 | 1.464 | **0.176** | 0.178 |
| Lasso | 2.572 | 2.079 | 0.270 | 0.245 |
| RSC | 2.644 | 2.341 | 0.239 | 0.215 |
| TabPFN | **1.441** | **1.231** | 0.179 | **0.169** |
| TabPFN-TS | 1.635 | 1.485 | 0.199 | 0.197 |
| Chronos | 1.509 | 1.366 | 0.202 | 0.201 |

Table 3. Median loss of each method under low and high levels of random and correlated noise.

dom and one correlated with the latent state. Both correlation structures have the same expected magnitude (see Appendix 1). Table 3 reports median MSE across these four settings. Noise magnitude scales the absolute error of all methods, while correlation structure has comparatively little effect. TabPFN achieves the lowest or near-lowest median MSE in most settings, with Simplex as the next best baseline (and best for low, correlated noise). We that foundation models exhibit noise robustness above or similar to the strongest traditional baselines.

### 4.3. Scaling with Panel Rank

Synthetic control typically assumes approximately low-rank structure in the panel matrix. Figure 3 shows results as a function of relative rank, defined as the panel's rank divided by the minimum of the sample and time dimensions. Robust SC remains are most competitive at the lowest relative ranks (0.04, 0.1, and 0.4), while TabPFN outperforms traditional methods at relative ranks of 0.1, 0.2, 0.25, 0.5, and 1.0. We find that foundation models offer substantive gains in high-rank regimes where classical low-rank assumptions are violated.

## 5. Discussion

### 5.1. Limitations

Our evaluation is conducted entirely on synthetic panels generated from state-space models. While this design lets us systematically vary structural assumptions such as linearity, noise, and rank, real-world panels may follow structures not well represented by our simulated data. We highlight opportunities to extend our evaluation below.

### 5.2. Future Work

**Adapting foundation models to synthetic control.** The foundation models we evaluate were pretrained for generic tabular or time-series prediction, not for the specific structure of synthetic control. The strong performance we observe despite this mismatch suggests both the promise of large-scale synthetic pretraining and opportunities to adapt or design models tailored to the synthetic control problem.

**Benchmarking improvements and practitioner guidance.** Synthetic control evaluations have historically relied on a small number of canonical case studies, such as Proposition 99 and German reunification, where the counterfactual is unobservable and rigorous comparison across methods is difficult. Our analysis represents a step towards more comprehensive evaluation, and we hope to expand our work with a suite of real data benchmarks in our final release, as well as considering semi-synthetic settings. Our results enable richer comparison of the data regimes in which different predictors are effective, and where each breaks down. Building tooling and guidance for practitioners to study when each regime applies in practice remains an important direction for future work.

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

# A. Appendix

## A.1. Parameter Sweep for Data Generation

We provide a more comprehensive description of our data generation process in Table 4. For the transition matrix, *orthogonal* refers to drawing a random Gaussian matrix $G \sim \mathcal{N}(0, I)^{d \times d}$ and taking the $Q$ factor of its QR decomposition. This places every eigenvalue exactly on the unit circle ($\rho = 1$), giving volume-preserving rotational dynamics. *Rescaled* instead draws $G$ and divides by its spectral radius before scaling by the target $\rho \in \{1.0, 0.8\}$. Eigenvalues are not constrained to the unit circle, so the same $\rho$ can produce a mixture of contracting, expanding, and oscillating modes. For the loading distribution, $\mathrm{Dir}_5$ draws each row of $H$ and $C$ from a symmetric Dirichlet with concentration $\kappa = 5$ (rows lie on the simplex with moderate concentration), while $\mathrm{Gauss}$ draws each entry iid $\mathcal{N}(0, 1)$ (signed and unconstrained). The noise variants control the structure of $Q$ and $R$: *diag* draws independent per-feature variances; *wishart-$\nu=d+1$* and *wishart-$\nu=10d$* draw correlated covariances around the same diagonal mean, with off-diagonal correlation scaling as $\sim 1/\sqrt{\nu}$ (so $\nu = d + 1$ gives strongly correlated noise and $\nu = 10d$ near-diagonal noise). The noise bin selects the variance band: low $= [0, 1]$, high $= [4, 5]$, with per-feature variances drawn iid uniformly within the chosen band.

*Table 4.* Data generation parameter sweep. A total of 16,200 parameter combinations, each with 20 independent samples generated. We perform a complete grid except for under-specified samples, where rank $d > min(n, T)$, resulting in a total of 324,000 SC panels evaluated.

| Parameter | Values / range | Description |
| --- | --- | --- |
| **Swept (23,040 parameter combinations)** | | |
| $\alpha = \beta$ | $\{0,\ 0.25,\ 0.5,\ 0.75,\ 1\}$ | Linear/nonlinear mixing coefficient. |
| Transition Matrix | $\{(\text{ortho}, \rho{=}1),\ (\text{rescaled}, \rho{=}1),\ (\text{rescaled}, \rho{=}0.8)\}$ | Transition matrix construction. |
| Loading Distribution | $\{\mathrm{Dir}_5,\ \mathrm{Gauss}\}$ | Observation matrix rows. |
| Noise Variant | $\{\text{diag},\ \text{wishart-}\nu{=}d{+}1,\ \text{wishart-}\nu{=}10d\}$ | Noise covariance structure. |
| Noise Level | $\{\text{low},\ \text{high}\}$ | Noise variance band. |
| $x_0$ Initial State Distribution | $\{\mathrm{Dir}_5,\ \mathrm{Gauss}\}$ | Initial latent state distribution. |
| $d$ (latent dim.) | $\{2,\ 5,\ 10,\ 20\}$ | Latent state dimension. |
| $n$ (donors) | $\{5,\ 10,\ 20,\ 50\}$ | Number of donor units. |
| $T_{\text{pre}}$ | $\{5,\ 10,\ 20,\ 50\}$ | Pre-intervention period length. |
| **Fixed across all parameter combinations** | | |
| $T_{\text{post}}$ | $T_{\text{pre}}$ | Post-intervention horizon. |
| burn-in | 5 latent steps | Burn-in length. |
| normalization | none | Donor-matrix normalization. |
| samples per parameter combination | 20 | Panels generated for each parameter combination. |

# B. Per-axis performance

We report mean and median MSE on $h \leq 5$ extrapolation across the three axes that drive most of the qualitative differences in our sweep: linearity, loading distribution, and relative rank. For each axis we report results on clean and noisy targets separately. CIs are 95% closed-form (mean: CLT; median: order-statistics binomial). The best baseline per row is shown in **bold**.

## B.1. Hyperparameter Tuning

*Table 5.* RSC and Lasso tuning procedures compared across data regimes. Median MSE vs. clean targets ($h \leq 5$); best variant per family (RSC, Lasso) in each row shown in **bold**; 95% CIs in grey.

| | RSC | RSC (CV) | RSC (silhouette) | Lasso | Lasso (CV) |
|---|---|---|---|---|---|
| *Linearity coefficient ($\alpha = \beta$)* | | | | | |
| 0.00 | 1.53 (1.50, 1.56) | 1.59 (1.56, 1.62) | **1.12** (1.10, 1.15) | 1.59 (1.55, 1.63) | **1.44** (1.41, 1.47) |
| 0.25 | 1.04 (1.02, 1.05) | 1.08 (1.06, 1.10) | **0.66** (0.65, 0.67) | 1.02 (0.99, 1.04) | **0.90** (0.88, 0.91) |
| 0.50 | 0.90 (0.89, 0.92) | 0.92 (0.91, 0.94) | **0.67** (0.65, 0.68) | 0.82 (0.79, 0.84) | **0.70** (0.69, 0.71) |
| 0.75 | 0.84 (0.83, 0.86) | 0.84 (0.83, 0.86) | **0.72** (0.70, 0.73) | 0.73 (0.71, 0.75) | **0.59** (0.58, 0.60) |
| 1.00 | 0.87 (0.85, 0.88) | 0.87 (0.85, 0.89) | **0.78** (0.77, 0.80) | 0.78 (0.76, 0.80) | **0.62** (0.61, 0.63) |
| *Relative rank $d/\min(n, T)$* | | | | | |
| 0.04 | 0.26 (0.24, 0.28) | 0.33 (0.31, 0.35) | **0.15** (0.14, 0.16) | 0.63 (0.53, 0.77) | **0.44** (0.42, 0.46) |
| 0.1 | 0.43 (0.42, 0.44) | 0.51 (0.50, 0.53) | **0.29** (0.28, 0.30) | 0.69 (0.65, 0.74) | **0.56** (0.55, 0.58) |
| 0.2 | 0.89 (0.87, 0.91) | 0.90 (0.87, 0.92) | **0.53** (0.52, 0.54) | 0.83 (0.80, 0.86) | **0.74** (0.73, 0.76) |
| 0.25 | 0.56 (0.55, 0.58) | 0.69 (0.67, 0.71) | **0.51** (0.49, 0.52) | 0.77 (0.72, 0.81) | **0.60** (0.59, 0.62) |
| 0.4 | 1.48 (1.45, 1.51) | 1.30 (1.27, 1.33) | **0.97** (0.95, 0.99) | 1.00 (0.98, 1.03) | **0.91** (0.90, 0.93) |
| 0.5 | 0.96 (0.94, 0.98) | 1.02 (1.00, 1.04) | **0.83** (0.82, 0.85) | 0.92 (0.90, 0.95) | **0.76** (0.75, 0.78) |
| 1 | 1.39 (1.36, 1.40) | 1.32 (1.30, 1.34) | **1.15** (1.13, 1.16) | 1.08 (1.06, 1.10) | **0.90** (0.89, 0.91) |

## B.2. Linearity ($\alpha = \beta$)

In Tables 6 and 7 we report performance by linearity coefficient $\alpha = \beta$, where $\alpha = 0$ is fully linear and $\alpha = 1$ is fully nonlinear. We find that Simplex performs the best in the mostly-linear regime ($\alpha \in \{0.25, 0.5\}$), while TabPFN performs better in nonlinear regimes ($\alpha \in \{0.75, 1.0\}$). The pattern is consistent across mean and median and across clean and noisy targets.

*Table 6.* Linearity coefficient ($\alpha = \beta$) – MSE (clean targets, $h \leq 5$). Each axis value spans two rows: Mean (top) and Median (bottom). Best per row in **bold**; 95% CIs in grey. RSC = silhouette-selected rank; Lasso = CV-selected $\lambda$ (best variant per family; see Table 5).

| $\alpha$ | Stat | Simplex | Lasso | RSC | TabPFN | TabPFN-TS | Chronos |
|---|---|---|---|---|---|---|---|
| 0.00 | Mean | 12.29 (10.79, 13.78) | 6.81 (6.55, 7.06) | 9.95 (9.06, 10.84) | **5.66** (5.46, 5.86) | 6.14 (5.96, 6.32) | 6.89 (6.70, 7.08) |
| | Median | **1.08** (1.06, 1.10) | 1.44 (1.41, 1.47) | 1.12 (1.10, 1.15) | 1.18 (1.16, 1.21) | 1.30 (1.27, 1.32) | 1.43 (1.41, 1.46) |
| 0.25 | Mean | **1.70** (1.68, 1.73) | 3.56 (3.46, 3.65) | 5.58 (4.05, 7.11) | 1.88 (1.84, 1.91) | 2.13 (2.10, 2.17) | 2.05 (2.02, 2.08) |
| | Median | **0.60** (0.59, 0.61) | 0.90 (0.88, 0.91) | 0.66 (0.65, 0.67) | 0.60 (0.59, 0.61) | 0.68 (0.67, 0.69) | 0.67 (0.66, 0.69) |
| 0.50 | Mean | **1.13** (1.11, 1.14) | 2.74 (2.64, 2.83) | 8.24 (4.51, 11.96) | 1.16 (1.14, 1.18) | 1.34 (1.32, 1.35) | 1.27 (1.25, 1.28) |
| | Median | 0.48 (0.47, 0.49) | 0.70 (0.69, 0.71) | 0.67 (0.65, 0.68) | **0.43** (0.42, 0.44) | 0.49 (0.48, 0.50) | 0.47 (0.46, 0.48) |
| 0.75 | Mean | 0.91 (0.90, 0.92) | 2.42 (2.27, 2.57) | 15.59 (9.28, 21.90) | **0.86** (0.84, 0.87) | 1.02 (1.00, 1.03) | 0.93 (0.92, 0.94) |
| | Median | 0.43 (0.43, 0.44) | 0.59 (0.58, 0.60) | 0.72 (0.70, 0.73) | **0.36** (0.36, 0.37) | 0.42 (0.41, 0.42) | 0.38 (0.38, 0.39) |
| 1.00 | Mean | 0.90 (0.89, 0.91) | 2.27 (2.20, 2.33) | 14.03 (6.36, 21.70) | **0.82** (0.81, 0.83) | 0.98 (0.97, 1.00) | 0.89 (0.88, 0.91) |
| | Median | 0.48 (0.47, 0.49) | 0.62 (0.61, 0.63) | 0.78 (0.77, 0.80) | **0.41** (0.40, 0.41) | 0.46 (0.46, 0.47) | 0.43 (0.42, 0.44) |

## B.3. Loading Distribution

In Tables 8 and 9 we report performance by loading distribution. Simplex performs the best on Dirichlet loadings, where the data-generating weights live close to the simplex, while TabPFN dominates on Gaussian loadings, which produce dense, sign-mixed weights.

*Table 7.* Linearity coefficient ($\alpha = \beta$) – MSE (noisy targets, $h \leq 5$). Each axis value spans two rows: Mean (top) and Median (bottom). Best per row in **bold**; 95% CIs in grey. RSC = silhouette-selected rank; Lasso = CV-selected $\lambda$ (best variant per family; see Table 5).

| $\alpha$ | Stat | Simplex | Lasso | RSC | TabPFN | TabPFN-TS | Chronos |
|---|---|---|---|---|---|---|---|
| 0.00 | Mean | 14.59 (13.10, 16.07) | 8.78 (8.52, 9.04) | 12.37 (11.48, 13.26) | **7.81** (7.61, 8.01) | 8.35 (8.17, 8.53) | 9.17 (8.97, 9.37) |
| | Median | 2.68 (2.63, 2.74) | 2.64 (2.59, 2.69) | 3.00 (2.94, 3.07) | **2.52** (2.47, 2.57) | 2.71 (2.66, 2.76) | 3.00 (2.93, 3.06) |
| 0.25 | Mean | **3.92** (3.88, 3.97) | 5.42 (5.32, 5.52) | 7.86 (6.31, 9.40) | 4.01 (3.96, 4.06) | 4.35 (4.30, 4.40) | 4.27 (4.23, 4.32) |
| | Median | 1.72 (1.69, 1.76) | 1.93 (1.89, 1.97) | 2.07 (2.03, 2.11) | **1.71** (1.67, 1.74) | 1.85 (1.81, 1.89) | 1.84 (1.80, 1.88) |
| 0.50 | Mean | 3.31 (3.28, 3.34) | 4.58 (4.47, 4.68) | 10.36 (6.60, 14.11) | **3.27** (3.24, 3.31) | 3.54 (3.51, 3.58) | 3.48 (3.45, 3.52) |
| | Median | 1.54 (1.51, 1.57) | 1.70 (1.67, 1.73) | 1.98 (1.94, 2.02) | **1.50** (1.47, 1.53) | 1.63 (1.60, 1.66) | 1.62 (1.59, 1.65) |
| 0.75 | Mean | 3.06 (3.04, 3.09) | 4.23 (4.08, 4.38) | 17.54 (11.25, 23.83) | **2.96** (2.93, 2.99) | 3.22 (3.19, 3.25) | 3.14 (3.11, 3.17) |
| | Median | 1.46 (1.43, 1.48) | 1.60 (1.57, 1.64) | 2.02 (1.98, 2.06) | **1.41** (1.39, 1.44) | 1.54 (1.51, 1.56) | 1.52 (1.49, 1.55) |
| 1.00 | Mean | 3.06 (3.03, 3.08) | 4.06 (3.99, 4.14) | 16.00 (8.33, 23.66) | **2.92** (2.90, 2.95) | 3.19 (3.15, 3.22) | 3.10 (3.08, 3.13) |
| | Median | 1.49 (1.46, 1.51) | 1.64 (1.61, 1.67) | 2.03 (1.99, 2.06) | **1.43** (1.41, 1.46) | 1.57 (1.54, 1.59) | 1.54 (1.52, 1.57) |

*Table 8.* Loading distribution – MSE (clean targets, $h \leq 5$). Each axis value spans two rows: Mean (top) and Median (bottom). Best per row in **bold**; 95% CIs in grey. RSC = silhouette-selected rank; Lasso = CV-selected $\lambda$ (best variant per family; see Table 5).

| Loading | Stat | Simplex | Lasso | RSC | TabPFN | TabPFN-TS | Chronos |
|---|---|---|---|---|---|---|---|
| Dir(5) | Mean | **0.84** (0.83, 0.85) | 2.57 (2.52, 2.62) | 9.87 (6.03, 13.70) | 1.16 (1.14, 1.17) | 1.29 (1.27, 1.30) | 1.42 (1.40, 1.44) |
| | Median | **0.35** (0.35, 0.36) | 0.63 (0.62, 0.64) | 0.51 (0.50, 0.51) | 0.39 (0.38, 0.39) | 0.44 (0.44, 0.45) | 0.44 (0.43, 0.44) |
| Gaussian | Mean | 5.92 (5.33, 6.52) | 4.54 (4.42, 4.67) | 11.49 (9.53, 13.44) | **2.99** (2.91, 3.07) | 3.36 (3.29, 3.43) | 3.39 (3.32, 3.47) |
| | Median | 0.84 (0.83, 0.85) | 0.97 (0.96, 0.98) | 1.14 (1.12, 1.15) | **0.66** (0.65, 0.66) | 0.75 (0.74, 0.76) | 0.70 (0.69, 0.70) |

### B.4. Relative Rank ($d/\min(n, T)$)

In Tables 10 and 11 we report performance by relative rank $d/\min(n, T)$. RSC performs the best at the deeply over-determined corner ($d/\min(n, T) = 0.04$), where its additional denoising provides benefits, while TabPFN dominates at every higher relative rank $d/\min(n, T) = 1$.

### B.5. Noise (magnitude and structure)

In Tables 12 and 13 we report performance by noise magnitude (low/high) and noise structure (random for diagonal $R$, correlated for Wishart $R$ pooling both concentration variants).

*Table 9.* Loading distribution – MSE (noisy targets, $h \leq 5$). Each axis value spans two rows: Mean (top) and Median (bottom). Best per row in **bold**; 95% CIs in grey. RSC = silhouette-selected rank; Lasso = CV-selected $\lambda$ (best variant per family; see Table 5).

| Loading | Stat | Simplex | Lasso | RSC | TabPFN | TabPFN-TS | Chronos |
|---|---|---|---|---|---|---|---|
| Dir(5) | Mean | **3.00** (2.98, 3.01) | 4.39 (4.34, 4.45) | 12.02 (8.19, 15.85) | 3.26 (3.23, 3.29) | 3.50 (3.48, 3.52) | 3.66 (3.64, 3.69) |
|  | Median | **1.38** (1.36, 1.39) | 1.64 (1.62, 1.66) | 1.85 (1.82, 1.88) | 1.46 (1.44, 1.47) | 1.58 (1.56, 1.60) | 1.64 (1.61, 1.66) |
| Gaussian | Mean | 8.18 (7.58, 8.78) | 6.43 (6.31, 6.56) | 13.63 (11.66, 15.59) | **5.13** (5.05, 5.21) | 5.56 (5.49, 5.64) | 5.61 (5.53, 5.69) |
|  | Median | 2.11 (2.08, 2.13) | 2.12 (2.09, 2.15) | 2.56 (2.53, 2.59) | **1.89** (1.86, 1.91) | 2.04 (2.02, 2.07) | 2.04 (2.01, 2.06) |

*Table 10.* Relative rank $d/\min(n, T)$ – MSE (clean targets, $h \leq 5$). Each axis value spans two rows: Mean (top) and Median (bottom). Best per row in **bold**; 95% CIs in grey. RSC = silhouette-selected rank; Lasso = CV-selected $\lambda$ (best variant per family; see Table 5).

| $d/\min(n,T)$ | Stat | Simplex | Lasso | RSC | TabPFN | TabPFN-TS | Chronos |
|---|---|---|---|---|---|---|---|
| 0.04 | Mean | 10.50 (4.73, 16.27) | 1.33 (1.27, 1.39) | 1.10 (0.92, 1.28) | 0.82 (0.79, 0.86) | **0.82** (0.79, 0.85) | 1.21 (0.94, 1.48) |
|  | Median | 0.27 (0.25, 0.29) | 0.44 (0.42, 0.46) | **0.15** (0.14, 0.16) | 0.34 (0.33, 0.36) | 0.35 (0.33, 0.36) | 0.33 (0.31, 0.34) |
| 0.1 | Mean | 3.11 (2.25, 3.96) | 1.87 (1.80, 1.94) | 2.27 (2.03, 2.52) | **0.97** (0.94, 1.00) | 1.03 (1.01, 1.05) | 1.25 (1.19, 1.30) |
|  | Median | 0.37 (0.36, 0.38) | 0.56 (0.55, 0.58) | **0.29** (0.28, 0.30) | 0.35 (0.34, 0.36) | 0.37 (0.36, 0.38) | 0.41 (0.39, 0.41) |
| 0.2 | Mean | 3.57 (2.52, 4.63) | 2.87 (2.74, 3.01) | 4.64 (3.84, 5.45) | **1.59** (1.51, 1.67) | 1.78 (1.72, 1.85) | 2.02 (1.94, 2.10) |
|  | Median | 0.49 (0.48, 0.50) | 0.74 (0.73, 0.76) | 0.53 (0.52, 0.54) | **0.48** (0.47, 0.49) | 0.55 (0.54, 0.56) | 0.55 (0.54, 0.56) |
| 0.25 | Mean | 1.81 (1.61, 2.02) | 2.12 (2.00, 2.23) | 3.30 (2.10, 4.50) | **1.07** (1.04, 1.10) | 1.21 (1.17, 1.25) | 1.28 (1.23, 1.32) |
|  | Median | 0.43 (0.42, 0.45) | 0.60 (0.59, 0.62) | 0.51 (0.49, 0.52) | **0.38** (0.37, 0.39) | 0.43 (0.42, 0.45) | 0.42 (0.41, 0.44) |
| 0.4 | Mean | 3.35 (2.98, 3.72) | 4.60 (4.36, 4.85) | 16.80 (10.46, 23.13) | **2.76** (2.61, 2.90) | 3.13 (3.00, 3.26) | 3.53 (3.38, 3.69) |
|  | Median | **0.64** (0.62, 0.65) | 0.91 (0.90, 0.93) | 0.97 (0.95, 0.99) | 0.64 (0.62, 0.65) | 0.75 (0.74, 0.77) | 0.73 (0.72, 0.75) |
| 0.5 | Mean | 2.49 (2.31, 2.68) | 3.08 (2.98, 3.19) | 8.65 (4.72, 12.58) | **1.64** (1.59, 1.68) | 1.91 (1.86, 1.96) | 1.96 (1.90, 2.02) |
|  | Median | 0.56 (0.55, 0.57) | 0.76 (0.75, 0.78) | 0.83 (0.82, 0.85) | **0.49** (0.48, 0.50) | 0.54 (0.53, 0.55) | 0.52 (0.51, 0.53) |
| 1 | Mean | 3.71 (3.10, 4.32) | 4.41 (4.29, 4.54) | 15.26 (10.19, 20.33) | **2.72** (2.63, 2.80) | 2.99 (2.92, 3.07) | 2.81 (2.74, 2.88) |
|  | Median | 0.68 (0.67, 0.69) | 0.90 (0.89, 0.91) | 1.15 (1.13, 1.16) | **0.57** (0.56, 0.58) | 0.66 (0.65, 0.67) | 0.60 (0.60, 0.61) |

*Table 11.* Relative rank $d/\min(n, T)$ – MSE (noisy targets, $h \leq 5$). Each axis value spans two rows: Mean (top) and Median (bottom). Best per row in **bold**; 95% CIs in grey. RSC = silhouette-selected rank; Lasso = CV-selected $\lambda$ (best variant per family; see Table 5).

| $d/\min(n,T)$ | Stat | Simplex | Lasso | RSC | TabPFN | TabPFN-TS | Chronos |
|---|---|---|---|---|---|---|---|
| 0.04 | Mean | 12.67 (6.91, 18.42) | 2.91 (2.82, 3.00) | 3.41 (3.21, 3.60) | **2.77** (2.69, 2.85) | 2.84 (2.76, 2.92) | 3.48 (3.19, 3.75) |
|  | Median | 1.37 (1.28, 1.45) | 1.40 (1.31, 1.46) | 1.49 (1.40, 1.60) | **1.32** (1.25, 1.41) | 1.36 (1.29, 1.45) | 1.49 (1.42, 1.57) |
| 0.1 | Mean | 5.34 (4.47, 6.22) | 3.54 (3.46, 3.62) | 4.50 (4.24, 4.77) | **3.01** (2.96, 3.05) | 3.16 (3.12, 3.21) | 3.54 (3.47, 3.61) |
|  | Median | 1.50 (1.45, 1.55) | 1.49 (1.44, 1.53) | 1.61 (1.55, 1.67) | **1.39** (1.34, 1.43) | 1.48 (1.44, 1.53) | 1.59 (1.55, 1.64) |
| 0.2 | Mean | 5.73 (4.67, 6.79) | 4.65 (4.50, 4.79) | 6.74 (5.93, 7.55) | **3.65** (3.56, 3.75) | 3.94 (3.86, 4.02) | 4.24 (4.15, 4.34) |
|  | Median | 1.57 (1.53, 1.61) | 1.72 (1.68, 1.76) | 1.79 (1.75, 1.83) | **1.52** (1.48, 1.56) | 1.64 (1.60, 1.68) | 1.73 (1.69, 1.77) |
| 0.25 | Mean | 3.94 (3.73, 4.15) | 3.82 (3.70, 3.94) | 5.41 (4.19, 6.63) | **3.09** (3.04, 3.15) | 3.31 (3.25, 3.37) | 3.46 (3.40, 3.53) |
|  | Median | 1.52 (1.47, 1.57) | 1.50 (1.45, 1.55) | 1.75 (1.68, 1.82) | **1.44** (1.40, 1.48) | 1.54 (1.49, 1.59) | 1.57 (1.51, 1.63) |
| 0.4 | Mean | 5.68 (5.31, 6.06) | 6.61 (6.36, 6.86) | 19.02 (12.71, 25.33) | **5.00** (4.85, 5.14) | 5.44 (5.31, 5.58) | 5.82 (5.66, 5.98) |
|  | Median | **1.80** (1.76, 1.83) | 2.10 (2.05, 2.15) | 2.41 (2.36, 2.45) | 1.88 (1.84, 1.91) | 2.06 (2.02, 2.10) | 2.08 (2.04, 2.13) |
| 0.5 | Mean | 4.66 (4.47, 4.85) | 4.86 (4.74, 4.97) | 10.69 (6.75, 14.63) | **3.71** (3.66, 3.77) | 4.10 (4.04, 4.16) | 4.12 (4.05, 4.19) |
|  | Median | 1.73 (1.70, 1.77) | 1.81 (1.77, 1.85) | 2.17 (2.12, 2.22) | **1.59** (1.57, 1.63) | 1.75 (1.72, 1.79) | 1.74 (1.70, 1.77) |
| 1 | Mean | 5.90 (5.29, 6.50) | 6.37 (6.24, 6.50) | 17.42 (12.35, 22.49) | **4.88** (4.79, 4.97) | 5.23 (5.15, 5.31) | 5.04 (4.96, 5.11) |
|  | Median | 1.84 (1.81, 1.86) | 2.08 (2.05, 2.12) | 2.58 (2.54, 2.62) | **1.80** (1.77, 1.83) | 1.94 (1.91, 1.97) | 1.93 (1.90, 1.96) |

*Table 12.* Noise (magnitude / structure) – MSE (clean targets, $h \leq 5$). Each row spans two lines: Mean (top) and Median (bottom). Best per row in **bold**; 95% CIs in grey. RSC = silhouette-selected rank; Lasso = CV-selected $\lambda$ (best variant per family; see Table 5).

| Noise (mag / struct) | Stat | Simplex | Lasso | RSC | TabPFN | TabPFN-TS | Chronos |
|---|---|---|---|---|---|---|---|
| Low / Random | Mean | 0.55 (0.53, 0.57) | 0.67 (0.64, 0.69) | 1.24 (0.80, 1.67) | **0.44** (0.42, 0.46) | 0.49 (0.47, 0.50) | 0.51 (0.50, 0.53) |
| | Median | 0.18 (0.18, 0.18) | 0.24 (0.24, 0.25) | 0.21 (0.21, 0.22) | **0.17** (0.17, 0.17) | 0.20 (0.20, 0.20) | 0.20 (0.20, 0.20) |
| Low / Correlated | Mean | 0.60 (0.55, 0.65) | 0.66 (0.65, 0.68) | 3.97 (-0.52, 8.46) | **0.43** (0.42, 0.44) | 0.48 (0.47, 0.49) | 0.52 (0.51, 0.53) |
| | Median | **0.18** (0.17, 0.18) | 0.27 (0.27, 0.27) | 0.24 (0.24, 0.24) | 0.18 (0.18, 0.18) | 0.20 (0.20, 0.20) | 0.20 (0.20, 0.20) |
| High / Random | Mean | 5.78 (5.08, 6.49) | 6.32 (6.14, 6.51) | 26.94 (18.72, 35.16) | **3.71** (3.57, 3.86) | 4.12 (4.00, 4.24) | 4.23 (4.10, 4.36) |
| | Median | 1.46 (1.45, 1.48) | 2.08 (2.05, 2.11) | 2.34 (2.30, 2.38) | **1.23** (1.22, 1.25) | 1.49 (1.47, 1.50) | 1.37 (1.35, 1.38) |
| High / Correlated | Mean | 6.39 (5.57, 7.21) | 6.52 (6.34, 6.69) | 13.97 (11.84, 16.11) | **3.72** (3.62, 3.81) | 4.18 (4.09, 4.27) | 4.33 (4.23, 4.42) |
| | Median | 1.49 (1.48, 1.50) | 2.57 (2.55, 2.60) | 2.64 (2.62, 2.67) | **1.44** (1.43, 1.45) | 1.64 (1.62, 1.65) | 1.51 (1.50, 1.52) |

*Table 13.* Noise (magnitude / structure) – MSE (noisy targets, $h \leq 5$). Each row spans two lines: Mean (top) and Median (bottom). Best per row in **bold**; 95% CIs in grey. RSC = silhouette-selected rank; Lasso = CV-selected $\lambda$ (best variant per family; see Table 5).

| Noise (mag / struct) | Stat | Simplex | Lasso | RSC | TabPFN | TabPFN-TS | Chronos |
|---|---|---|---|---|---|---|---|
| Low / Random | Mean | 1.05 (1.03, 1.07) | 1.16 (1.14, 1.18) | 1.73 (1.29, 2.17) | **0.94** (0.92, 0.96) | 0.98 (0.97, 1.00) | 1.01 (0.99, 1.02) |
| | Median | 0.60 (0.60, 0.61) | 0.68 (0.67, 0.68) | 0.68 (0.67, 0.69) | **0.59** (0.58, 0.59) | 0.62 (0.61, 0.63) | 0.63 (0.62, 0.64) |
| Low / Correlated | Mean | 1.02 (0.97, 1.07) | 0.99 (0.98, 1.01) | 4.38 (-0.10, 8.87) | **0.82** (0.81, 0.83) | 0.90 (0.89, 0.91) | 0.94 (0.93, 0.96) |
| | Median | 0.52 (0.52, 0.53) | 0.55 (0.55, 0.56) | 0.62 (0.61, 0.62) | **0.49** (0.49, 0.50) | 0.55 (0.54, 0.55) | 0.56 (0.56, 0.57) |
| High / Random | Mean | 10.31 (9.59, 11.02) | 10.80 (10.61, 10.99) | 31.55 (23.33, 39.77) | **8.18** (8.03, 8.33) | 8.60 (8.47, 8.73) | 8.71 (8.57, 8.85) |
| | Median | 5.82 (5.77, 5.86) | 6.67 (6.61, 6.73) | 7.27 (7.20, 7.33) | **5.66** (5.62, 5.71) | 6.01 (5.96, 6.05) | 5.87 (5.82, 5.91) |
| High / Correlated | Mean | 10.07 (9.25, 10.88) | 9.27 (9.09, 9.45) | 17.45 (15.31, 19.59) | **7.21** (7.11, 7.31) | 7.89 (7.80, 7.99) | 8.10 (8.00, 8.20) |
| | Median | 4.84 (4.81, 4.87) | 5.25 (5.22, 5.29) | 6.26 (6.23, 6.31) | **4.71** (4.68, 4.74) | 5.16 (5.13, 5.20) | 5.19 (5.16, 5.22) |

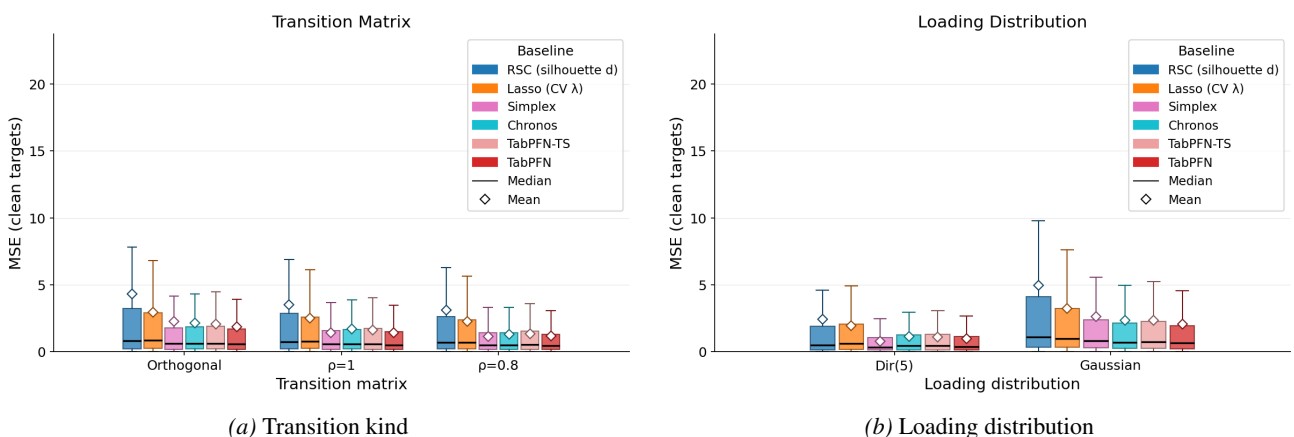

*(a)* Transition kind  *(b)* Loading distribution

*Figure 4.* Transition and loading ablations.

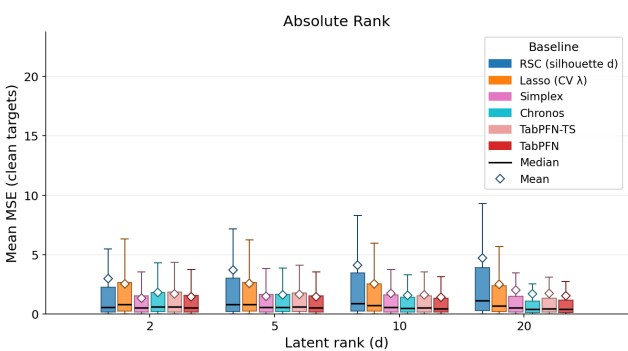

*Figure 5.* Absolute rank ($d$).

### B.6. Additional Axis Plots (Clean Targets)

We provide further breakdowns of our results according to transition matrix structure and latent loading distribution in Figure 4. We find that performance is consistent under parameterizations of the transition matrix. We find that when the latent loading distribution is sparse (Dirichlet distribution with $\kappa = 5$), simplex achieves the best performance, and when the latent loading distribution is dense (Gaussian), the foundation models achieve better performance.

We plot results according to absolute rank in Figure 5, and by the number of donors and time steps in Figure 6. We find that in the lowest absolute rank setting (2), simplex performs the best, and foundation models outperform linear methods in high rank settings. We find that method ranking is consistent under increasing number of donors, and more varied under increasing time steps.

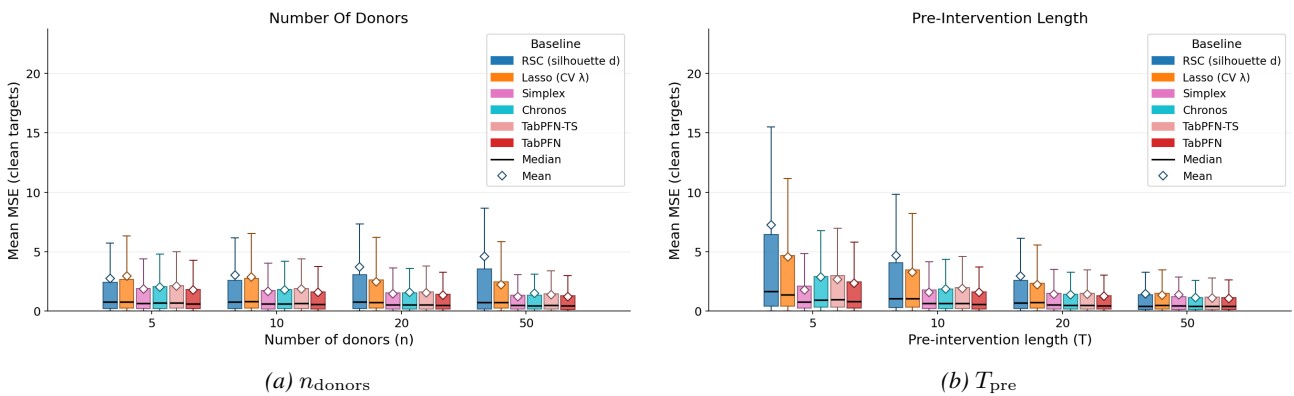

*(a)* $n_{\text{donors}}$                    *(b)* $T_{\text{pre}}$

*Figure 6.* Donor count and pre-period length ablations.

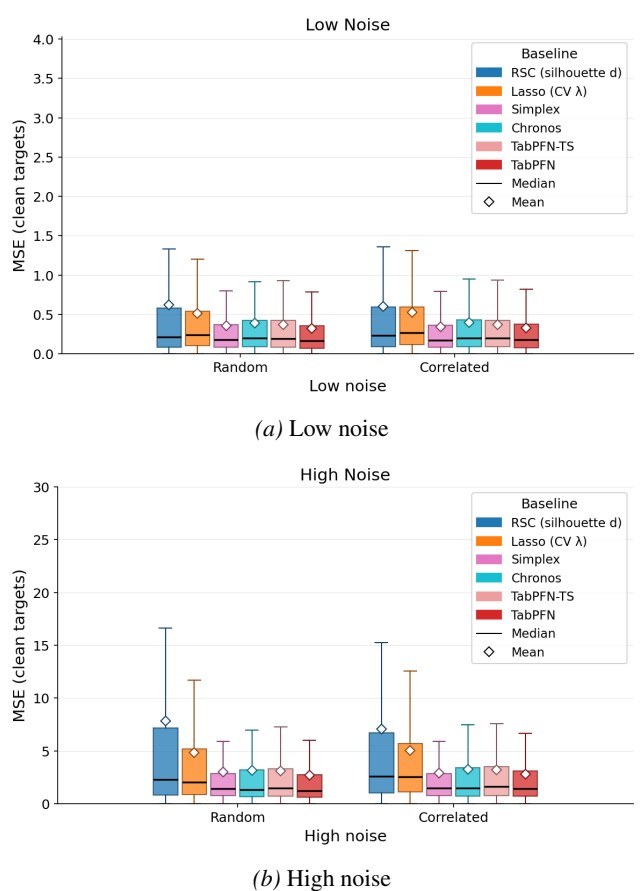

*(a)* Low noise

*(b)* High noise

*Figure 7*

## B.7. Plots Compared to Noisy Targets

Below, we provide replications of our results when compared to noisy targets rather than the clean underlying signal.

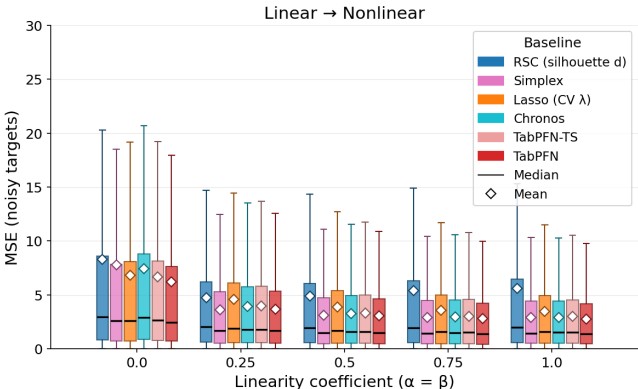

*Figure 8.* Linearity ($\alpha$).

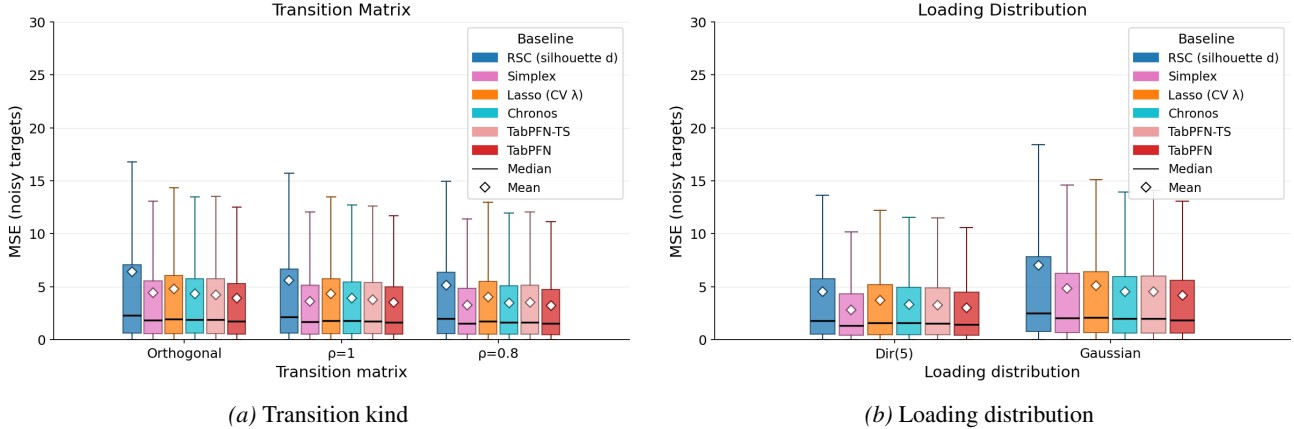

*(a)* Transition kind                    *(b)* Loading distribution

*Figure 9.* Transition and loading ablations.

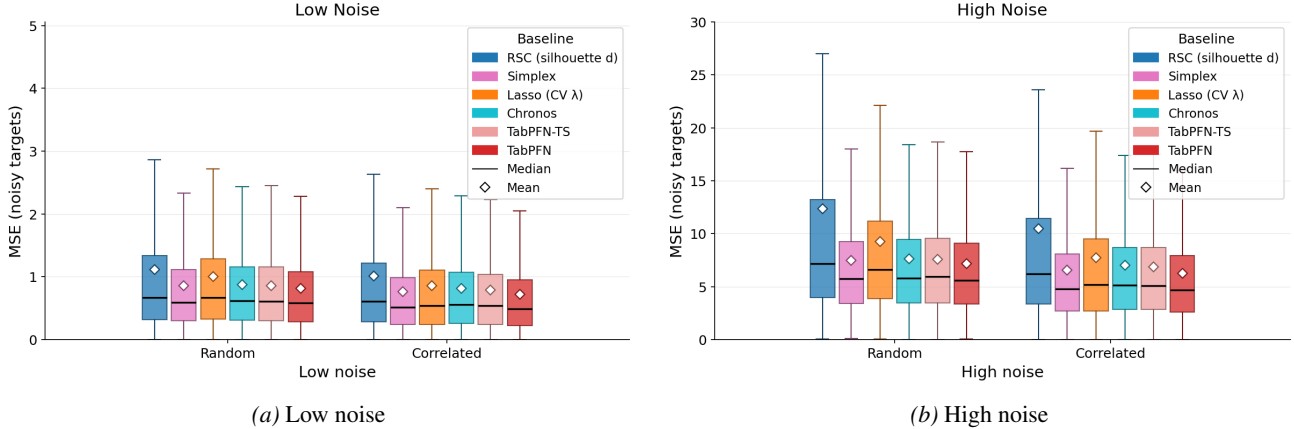

*(a)* Low noise

*(b)* High noise

*Figure 10.* Noise-structure ablations.

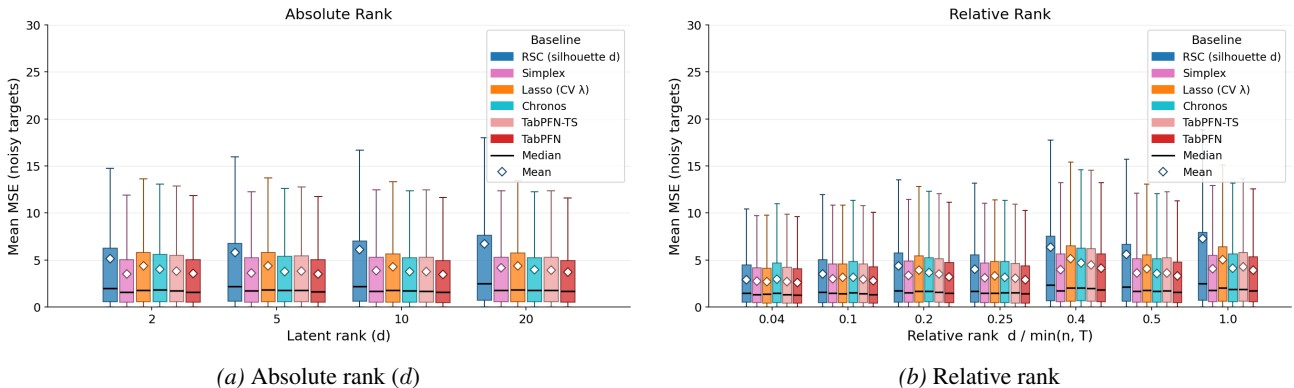

*(a)* Absolute rank (*d*)

*(b)* Relative rank

*Figure 11.* Rank ablations.

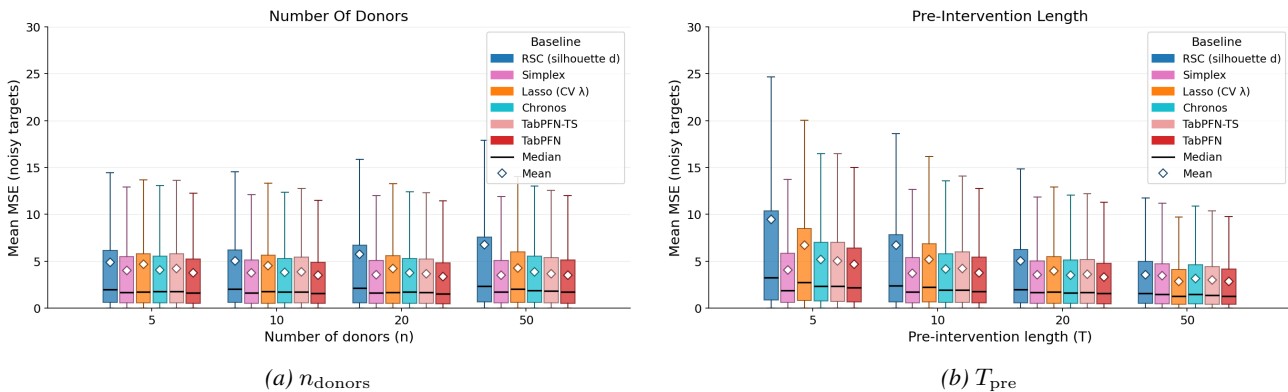

*(a)* $n_{\mathrm{donors}}$

*(b)* $T_{\mathrm{pre}}$

*Figure 12.* Donor count and pre-period length ablations.

