# OpenReview forum: "SCBench: A Testbed for Causal Inference with Time Series Panel Data"
_ICML.cc/2026/Workshop/FMSD — FMSD @ ICML 2026 Poster_

### Official Review · Reviewer_ezAu · 2026-05-12
**Review for "SCBench: A Testbed for Causal Inference with Time Series Panel Data"**

**Rating:** 6
**Confidence:** 3

**Review:**

### Summary

This paper studies whether foundation models can be useful for synthetic control. The authors build a large-scale synthetic benchmark with over 300K simulated panel datasets and compare traditional synthetic control methods with foundation models such as TabPFN, TabPFN-TS, and Chronos. The results show that foundation models are promising in nonlinear and high-rank regimes, while classical linear methods remain strong in low-rank or near-linear settings.

### Strengths

The paper extends the application scope of tabular and time series foundation models to synthetic control, which is an interesting direction.

The comparison between classical SC baselines and foundation models is valuable, especially because it highlights both the promise and the limitations of current foundation models in causal inference with panel data.

### Areas for Improvement

As the authors also acknowledge, the evaluation is entirely based on synthetic panels generated from state space models. Real world panel data may have structures that are not well captured by the current simulator, so experiments on real datasets are needed to better validate the practical relevance of the conclusions.

The paper should also consider evaluating more recent and stronger foundation models. Since the capabilities of foundation models are improving rapidly, the conclusions may depend on the specific model generation used in the experiments.

Finally, the paper could provide more guidance on how practitioners should choose between classical SC methods and foundation model methods in real applications.

---

### Official Review · Reviewer_UWz6 · 2026-05-19
**SCBench**

**Rating:** 5
**Confidence:** 4

**Review:**

## Summary

This paper presents SCBench, a large-scale empirical study comparing three foundation models (TabPFN, TabPFN-TS, Chronos) against three traditional synthetic control (SC) baselines (Robust SC, Lasso, Simplex) on over 300K simulated panel datasets.

## Strengths

- **Useful contribution**: A 300K-panel benchmark for synthetic control is genuinely novel. SC evaluations have historically been limited to a small number of canonical case studies (Proposition 99, German reunification) where ground truth is unobservable. The testbed lowers the barrier for systematic comparison.
- **Clear research question**: "When do foundation models help in synthetic control?" is well-scoped and timely given the rise of TabPFN/Chronos.
- **Honest, nuanced findings**: The authors don't oversell foundation models. The finding that Simplex remains the strongest reliable baseline is reported clearly, and the regime decomposition (Tables 2, 5-12) gives a clear picture of where each method wins.
- **Comprehensive ablation axes**: Linearity, noise magnitude × structure, transition matrix type, loading distribution, absolute rank, relative rank, donor count, pre-period length are systematically varied and reported in the appendix.
- **Both clean and noisy targets evaluated**: Appendix B.6 replicates all results against noisy targets, addressing the concern that good fits to clean signal don't necessarily indicate good fits to observed data.


## Areas for Improvement

- **Methodological contribution is thin**: The paper applies existing methods to a benchmark and reports results. The methods are not adapted for SC (TabPFN is treated as a regressor with donors as features; Chronos uses donors as known future covariates). This is acknowledged in Section 5.2 as future work, but it limits the depth of the contribution.
- **State-space prior is narrow**: The data-generating process is a particular form of state-space model with $\tanh$ nonlinearity. Real economic panels often exhibit (i) trends, seasonality, and regime changes; (ii) heterogeneous noise across donors; (iii) outliers and structural breaks; (iv) heterogeneous treatment timing. None of these are represented in the testbed. The limitations section acknowledges synthetic-only evaluation but the gap is larger than just "synthetic vs. real."
- **No real-data sanity check**: Even one real benchmark (e.g., Proposition 99, German reunification, Basque Country) where rough agreement with published estimates would corroborate the simulator-based ranking would strengthen the contribution.
- **No comparison to Augmented SC, Generalized SC, or Time-Aware SC**: The latter (Rho et al. 2026) is cited as relaxing constant donor-target relationships but not compared. Augmented SC (Ben-Michael et al. 2021) is a major recent baseline not included.
- **Missing recent forecasting FMs**: The Chronos family has evolved rapidly; what version of Chronos is used? Recent models like Moirai or TimesFM might also be relevant.
- **$T_{\text{post}} = T_{\text{pre}}$ is constrained**: Forecast horizons matching pre-period length is a strong restriction; in practice, post-intervention windows are often much longer than pre-period.
- **Concurrent work overlap**: Illick et al. (2026) is cited as concurrent and "explores approaches for adapting time-series foundation models for SC." This needs more discussion of how SCBench complements vs. duplicates that work.
- **Statistical significance / multiple comparisons**: With 6 methods × ~10 ablation axes × multiple settings, there are many comparisons. Some adjustment or explicit statement of comparison strategy would help.
- **No characterization of computational cost**: For practitioners, foundation model inference cost (especially Chronos) may be relevant. Median time per panel by method would be useful.
- **Figure 1 schematic is helpful but Figures 2-3 boxplots are dense**: Median markers and quartile ranges run together. Consider a cleaner visualization (e.g., line plots of mean MSE vs. linearity coefficient with error bands).


## Detailed Comments

1. The $\alpha = \beta$ constraint is convenient but artificial, it conflates latent-dynamics linearity with observation-equation linearity. Real systems can have linear dynamics with nonlinear observation (or vice versa). What happens at $\alpha \neq \beta$?
2. The TabPFN application is described briefly: donor units as features, time steps as rows, target trajectory as regression target. This treats temporal structure as exchangeable across rows. Does TabPFN-TS's positional encoding make any difference? The fact that vanilla TabPFN slightly outperforms TabPFN-TS is interesting and somewhat surprising.
3. For Chronos, donors are passed as "known future covariates." How does the model treat the panel structure? Some sensitivity to context length / chunking would be useful.
4. The "relative rank $d/\min(n, T)$" axis is interesting but it implicitly ties together three things: latent dimension, donor count, and pre-period length. Decomposing the effect cleanly is hard. Have you tried regressing MSE on each separately?
5. The result that Simplex is best in the $\alpha = 0$ (fully linear) setting is surprising given that Simplex restricts to the unit simplex. This is a *constraint*, not the maximum-likelihood estimator under a linear model. Is this because the simplex constraint acts as regularization in the low-data regime?
6. Could you report results when treatments truly change the data (i.e., post-intervention trajectories evaluated against counterfactuals from the *same* SCM)? Currently the "intervention" is implicit and the predictor is evaluated on extrapolation, which is a forecasting problem, not strictly a causal one.
7. The noise covariance with Wishart-($d+1$) vs. Wishart-($10d$) is creative but it's not clear which is more realistic for typical panel data applications. A brief justification would help.
8. The "TabPFN achieves the lowest mean and median error overall" claim: Given the variance and that 95% CIs overlap with Simplex on multiple key rows, this should be softened or replaced with a rank-aggregation argument.
9. The release of the testbed is mentioned ("anonymized"): Assuming this is released, please ensure the data-generation code, exact hyperparameters, and method wrappers are all reproducible.

## Justification of Score

This is a useful empirical study with an honest, nuanced characterization of where foundation models help vs. don't on a structured panel-data task. The 300K benchmark is a genuine contribution to a field where evaluation has historically been ad hoc. However, the methodological contribution is modest (no method adaptation, just benchmarking), the data-generating process is narrow, real data is absent, and several recent SC baselines are missing.

---

### Official Review · Reviewer_DjYU · 2026-05-20
**Synthetic Control benchmarked as a covariate-informed forecasting task using Foundations Models**

**Rating:** 6
**Confidence:** 4

**Review:**

**Summary**

The paper focuses on Synthetic Control, which aims to predict the counterfactual trajectory of a target series using similar covariate series. Typically, a model is fit on past values to capture the relationship between target and covariates, and then the deviation to the prediction corresponds to the treatment effect.

The authors propose to evaluate pretrained foundation models on this task. They generate a large panel of synthetic tasks using a state space model with varying parameters. Notably, the degree of linearity and the relative rank can be changed flexibly.

Results show that state-of-the-art fondation models adapt well to this task. In particular, TabPFN outperforms their strongest baseline (simplex method) in non-linear and mid to high rank settings.

**Strengths**

The synthetic generation process enables analysis of varying degree of linearity and dependence between the variables, and the authors have performed a large parameter sweep. They have also tested multiple state-of-the-art foundation models.

**Areas for improvement**

The experiment does not particularly evaluate synthetic control, but rather the general task of covariate-informed forecasting, on short horizons. Also, the experiment is purely synthetic and relying on one specific generator, which may not be sufficient to model realistic scenarios encounted in synthetic control.

**Detailed comments**

The goal of synthetic control is to estimate a treatment effect. Yet here, what is really evaluated is simply the capability of the models to make an accurate forecast of the horizon (which is notably short: only 5 time steps) by using the covariates. The authors could have compared to the FMs without the covariates (to evaluate the capability of the model to efficiently use the information provided by the covariates), and perhaps simulated an actual “treatment”.

Also, I wonder why TabPFN-TS Chronos perform worse than TabPFN, which is not designed for time series. What happens on longer horizons?

Example predictions on real-world data would have been interesting, e.g on actual econometric series.

Comments on form / typos:

- line 73 second column “appraoches” → approaches
- line 152 first column “all methods are fit” → fitted
- line 154 first column, is it T=Tpre ? why two notations?
- notation for Tpost = [1..5] is not very rigourous. Use \\llbracket or [1:5] or {1, \dots, 5 \} ?
- Why is 2.1. Data Generation Process in the related works section?

    Is it inspired from another paper? If yes, which one?

- “clean” data/targets is not defined. Is it the noiseless data? Why two different terms?